# Early-Onset Colorectal Cancer: A Review of Current Insights and a Call for Action

**DOI:** 10.3390/biomedicines13071572

**Published:** 2025-06-26

**Authors:** Shaina Ailawadi, David C. Kaelber, Abbinaya Elangovan

**Affiliations:** 1Department of Internal Medicine, University Hospitals Cleveland Medical Center, Case Western Reserve University, Cleveland, OH 44106, USA; 2Center for Clinical Informatics Research and Education, The MetroHealth System and the Departments of Internal Medicine, Pediatrics, and Population and Quantitative Health Sciences, Case Western Reserve University, Cleveland, OH 44106, USA; dkaelber@metrohealth.org; 3Division of Gastroenterology and Hepatology, Carilion Clinic-Virginia Tech Carilion School of Medicine, Roanoke, VA 24016, USA; aelangovan@carilionclinic.org

**Keywords:** early-onset colorectal cancer, colorectal cancer, epidemiology, microsatellite instability, modifiable risk factors

## Abstract

While the incidence of colorectal cancer (CRC) in patients above 50 years old has been decreasing over the last several decades, the incidence of CRC among younger patients has been increasing. Early-onset CRC (EO-CRC) is known to exhibit distinct characteristics including specific genetic mutations and modifiable risk factors, which warrants tailored screening and management approaches. This review synthesizes current insights on EO-CRC’s epidemiological, genetic, molecular, and clinical features, as well as a call for action for future research on prevention and clinical management.

## 1. Introduction

Colorectal cancer (CRC) is the third most common malignancy globally and the most common gastrointestinal cancer, comprising approximately 1.8 million new cancer diagnoses and over 880,000 deaths in 2018 [1]. However, the incidence of CRC among older people has been gradually decreasing globally over the last several decades, as a result of improved screening [2]. Paradoxically, over the past few decades, the incidence of early-onset CRC (EO-CRC) in adults younger than 50 years of age has been significantly increasing in both men and women globally [3,4]. Those with EO-CRC typically present with more advanced disease stages at their diagnosis, have more aggressive tumor features, and have greater ramifications in terms of life-years lost compared to those diagnosed with colorectal cancer at an older age. Additionally, patients with EO-CRC have unique features including genetic conditions such as Lynch syndrome and familiar adenomatous polyposis. This rising trend of EO-CRC creates a need to raise awareness and determine a diagnostic threshold in this younger population. 

Because a clear consensus to screen the younger population is lacking, it is necessary to explore current evidence on EO-CRC comprehensively. Our review aims to present the current knowledge and insights on EO-CRC’s epidemiological trends, genetic manifestations, molecular characteristics, and clinical features. Additionally, we discuss a call for action with the goal of highlighting areas of future research on the prevention and clinical management of EO-CRC. We hypothesize that the rapidly increasing incidence of EO-CRC will be a culmination of genetic factors, environmental exposures, and lifestyle which are unique to development of colorectal cancer in the younger population.

## 2. Epidemiology

Current data has shown an epidemiological shift in CRC incidence with a decrease of 3.1% per year among people 50 years and above. However, the global incidence of EO-CRC has been increasing in both men and women. The average annual percentage change in the incidence of EO-CRC from 2008 to 2012 was 4.0% in New Zealand, 2.8% in Canada, and 2.2% in the United States of America [5]. Further, in the United States, the incidence of EO-CRC has been increasing since the mid-1990s, with the age-adjusted incidence increasing from 5.9 to 8.4 cases per 100,000 persons between 2000 and 2017 [5]. Similarly, the incidence of EO-CRC has been rising among European countries, with the incidence increasing to up to 7.9% in individuals aged 20–49 [6]. In the past, it was unclear if EO-CRC displayed similar trends in lower-income countries including Asia and Africa due to limited studies [4]. But a recent 2025 population-based study found that the rise in EO-CRC is no longer confined to high-income western countries, but also extends to developing countries in eastern Europe, Asia, and Latin America and the Caribbean, leading to necessary attention to be brought to investigating and elucidating characteristics of EO-CRC [7]. It is estimated that in the next decade, 25% of rectal cancers and 10–12% of CRC will be diagnosed in individuals younger than 50 years [8].

The upward trend in the incidence of EO-CRC in most Western countries since the 1980s is hypothesized to reflect a phenomenon called the “birth cohort effect” in which temporal changes among risk factors may have differential impacts on each birth cohort, with the increased risk being perpetuated to a later time [9]. These trends are notable among Generation X (birth years of 1965–1980) who experienced an initial increase in EO-CRC, then a subsequent increased rate of 1.22-fold among individuals born in 1965–1969, 1.58-fold among individuals born 1975–1979, and further increases across successive generations [10]. This birth cohort shift may reflect secular trends in risk factor exposures during early life among the prenatal to adolescent period, compared to adulthood, causing a delayed effect on CRC incidence. The birth cohort effect is evident on a global scale, despite differences in population age, screening programs, and diagnostic strategies across the world. As our population ages, it is anticipated that rates of CRC will continue to increase in the future and high-risk birth cohorts become older [10].

## 3. Risk Factors

It is known that CRC stems from an interaction of genetic and environmental factors. The majority of CRCs are sporadic (70%), a small proportion of CRC cases are associated with inherited syndromes (5%), and the remaining (25%) have an associated hereditary component, known as familial CRC, that has not been well established [2]. Based on current literature, both modifiable and non-modifiable factors may contribute to EO-CRC.

### 3.1. Modifiable Risk Factors

The risk factors associated with EO-CRC identified in the case–control studies include a weight gain of ≥5 kg (within the 5-year period preceding colonoscopy), a high intake of alcohol or processed meat, and inflammatory bowel disease. Factors associated with a decreased risk of EO-CRC include aspirin use and a high intake of vegetables, citrus fruits, fish, β-carotene, vitamin C, vitamin E or folate [11,12].

Further, it has been established that a high intake of processed foods and high-glycemic foods may foster a favorable environment for colon proliferation. A prospective study in 2023 by Zheng et al. analyzed diet quality in >29,000 women between 25 and 42 years of age and found a positive relationship between Western diet and colorectal cancer diagnosed under 50 years of age (multivariable-adjusted odds ratio (OR) of 1.38 (95% CI: 1.13, 1.68)) [13]. There was also an inverse relationship between healthier dietary patterns including dietary approaches to stop hypertension (DASH) and the Mediterranean diet and colorectal adenocarcinoma before 50 years of age [13].

Obesity at an early age can increase CRC risk. From a meta-analysis by Hidayat et al., higher body adiposity at an early age is associated with a higher risk of colon cancer in men and women. However, this was not true to rectal cancer [14]. This correlation is hypothesized to be an effect of proinflammatory cytokines produced by the adipose tissue and chronic exposure to hyperinsulinemia and insulin-like growth factor 1 (IGF-1), which may contribute to carcinogenesis. Further, the protective role of physical activity has been described in different studies demonstrating an inverse relationship between physical activity and risk of colon cancer. This phenomenon could be due to decreased inflammation, reduced intestinal transit time, a reduction in IGF-I levels, and modulated immune function, which have been associated with physical activity [2].

Both alcohol and tobacco are independent risk factors of CRC, with a recent study showing a dose-dependent relationship [15]. It has been proposed that carcinogens associated with cigarette smoke including nitrosamines, benzene, heterocyclic amines, and polycyclic aromatic hydrocarbons reach the colorectal mucosa by direct ingestion and may have a significant carcinogenic impact on both the colon and the rectum [15]. There is increasing evidence that smoking is strongly associated with the microsatellite instability-high, CpG island methylator phenotype-positive, and B-Raf protein encoding gene (*BRAF*) mutation-positive subtypes of CRC, implying that epigenetic modification may be functionally involved in smoking-related colorectal carcinogenesis [16]. It is plausible that these mechanisms may impact EO-CRC as well as later-onset CRC, which has been supported by the consistent association of smoking and CRC [16].

### 3.2. Non-Modifiable Risk Factors

The most important risk factor for CRC is family history, as having a first-degree relatives (FDRs) with CRC diagnosed under the age of 50 increases the risk of developing CRC by more than two- to four-fold [8]. Additionally, family history of advanced adenomas has also been found to increase CRC risk [9]. Next, patients with familial adenomatous polyposis (FAP) have nearly 100% risk of CRC, and those with attenuated familial adenomatous polyposis have close to 69% risk of CRC [8]. In both syndromes, the APC gene is mutated at chromosome 5q21. Other known conditions that increase the risk for EO-CRC include Lynch syndrome, which is associated with 2–4% of all CRC as well as other malignancies [8]. Despite rigorous colonoscopy surveillance, it has been reported that approximately 1.2 million Americans affected by Lynch syndrome are undiagnosed [8,17]. Thus, the American Society of Clinical Oncology and the National Comprehensive Cancer Network (NCCN) have recommended universal screening for Lynch syndrome in CRC to further improve detection rates among this high-risk population. Further, patients with inflammatory bowel disease are at increased risk for EO-CRC due to long-term inflammation. The highest risk for EO-CRC is associated with pancolitis, long duration of colitis and/or a concomitant primary sclerosing cholangitis diagnosis. Hence, surveillance colonoscopy is recommended every 1–3 years after 8–10 years of diagnosis in ulcerative colitis, IBD (unclassified) extending beyond the rectum, and Crohn’s disease with involvement of at least one-third of the colon. PSC patients are recommended to undergo annual surveillance colonoscopy [18].

## 4. Clinical and Histopathological Characteristics of EO-CRC

### 4.1. EO-CRC Clinical Presentation and Diagnostic Delays

The increasing prevalence of EO-CRC has prompted significant research about its underlying mechanisms and characteristics, showing that there may be fundamental differences between EO-CRC and late-onset CRC (LO-CRC). Previous studies have highlighted the differences between EO-CRC and LO-CRC in terms of their clinical presentation, tumor location, and histological features, highlighting that EO-CRC could be a distinct entity from late-onset CRC. It has been established that EO-CRC is associated with abdominal pain, rectal bleeding, weight loss, anemia, reduced appetite, and changes in bowel habits, as well as a higher likelihood of presenting with bowel obstruction. However, the delay in diagnosis of EO-CRC is alarming, with one series showing an average delay of four to six months and with case reports noting a two-year delay, which could be due to lack of symptom awareness [2,8]. In a recent meta-analysis of 39 studies it was found that younger adults experience longer delays compared to older individuals, particularly before diagnosis [19]. To complement this, several studies consistently demonstrated that younger patients present more often with stage III or IV disease [2]. One multicenter retrospective study found that 61.2% of EO-CRC compared to 44.5% of older patients presented metastatic disease at first diagnosis [2]. This could be related to both patient and clinician factors including the lack of screening programs for younger patients, non-specific presenting symptoms, lack of awareness of CRC, misdiagnosis such as hemorrhoidal disease as well as the aggressive histopathological characteristics of EO-CRC, and its potential inheritance which may predispose young patients to accelerated carcinogenesis. However, there was no evidence that younger adults experience longer delays from diagnosis to treatment, and in fact, four studies found younger adults had shorter times from diagnosis to treatment [20].

### 4.2. EO-CRC Molecular and Histopathological Features

Early-onset colorectal cancer (EO-CRC) exhibits unique molecular and histopathological features that differentiate it from CRC diagnosed in older populations. These distinctions contribute to its often more aggressive presentation and have significant implications for diagnosis and treatment. In a recent meta-analysis of 149 articles, among patients with EO-CRC, there is a higher prevalence of mutations in several oncogenes linked to mortality and poor therapeutic response, including *KRAS*, *BRAF*, and *NRAS,* compared to individuals with late-onset disease (Table 1). These genes encode proteins which act downstream of the epidermal growth factor receptor (EGFR) and activate Mek/Erk signaling. These oncogenes are established negative predictive markers for EGFR inhibition in metastatic CRC and are associated with inferior survival outcomes, regardless of tumor staging. EO-CRC is also associated with a higher prevalence of potentially harmful mutations in *TP53* and *PTEN*. Loss of PTEN activity has been linked to resistance to EGFR inhibition in cases of metastatic CRC [15]. Furthermore, a significantly higher prevalence of TP53 mutations in EO-CRC can lead to the loss of p53 tumor suppression and pro-tumorigenic gain-of-function effects which can accelerate colonic cell proliferation, angiogenesis, and metastasis [20,21].

In addition to its distinct molecular profile, EO-CRC is associated with aggressive histological subtypes including poorly differentiated tumors, mucinous carcinomas, and signet ring cell carcinomas [20,21]. While signet ring carcinomas comprise only 1% of all CRCs, they constitute 2–3% of early-onset tumors and are associated with a high risk of mortality and recurrence.

The unique aggressive tumor presentation, distinct oncogenetics, and histological manifestations of EO-CRC highlight the need for close attention and differentiation in its diagnosis and treatment practices compared to CRC diagnosed later in life. Understanding these characteristics is crucial for developing tailored therapeutic strategies and improving outcomes for younger patients with CRC.

## 5. Screening in Early-Onset Colorectal Cancer

Colorectal cancers diagnosed during screening prior to symptom development have shown significantly improved outcomes. Cancers detected in screening showed a 74% reduction in CRC-related mortality compared to cancers diagnosed after symptom onset [23]. In addition to survival benefit, studies have also found that earlier screening at the age of 45 years old and even 40 years old is more cost-effective than conventional screening. Data projections from Canada have shown that screening from a younger age resulted in 18,135 fewer CRC cases, 7988 fewer cases of CRC-related mortality, and 150,373 quality-adjusted life years when screened at 45 years old compared to 50 years old [24].

Current guidelines by major organizations in the United States of America including the National Comprehensive Cancer Network, US Multi Society Task Force on Colorectal Cancer, and US Preventative Services Task Force (USPSTF) have recommended CRC screening from 45 years of age among average-risk adults. Earlier screening is recommended for individuals with at least one first-degree relative with CRC before the age of 60 years old or two or more first-degree relatives diagnosed at any age. A population-based prospective study on EO-CRC among newly diagnosed patients with invasive CRC from the Ohio CRC Prevention Initiative assessed that with early screening, at least 16% of patients could be diagnosed earlier [25]. The Delphi Initiative for Early-Onset Colorectal Cancer (DIRECt) guidelines, which are the first consensus recommendations on EO-CRC, recommend that assessing CRC risk, workup of symptoms and colonoscopy be performed within 30 days of presentation [26]. Further, in 2018, Abualkhair et al. investigated the incidence of CRC in 1-year age increments using a large population database. The study showed an increase in the incidence of CRC from 49 to 50 years (46.1% increase per 100,000) with an IRR of 1.46 (95% CI, 1.42 to 1.51), which supports the argument for earlier screening for CRC. Close adherence to CRC screening especially in symptomatic younger patients has been shown to reduce substantial burden of undetected cancers among younger adults who would not normally undergo screening at 50 years [27].

Barriers to early screening implementation may include lack of awareness among younger populations, financial barriers and insurance coverage gaps associated with early screening practices, and lack of awareness among physicians. Risk and benefits of EO-CRC screening should be weighed carefully and further consensus regarding evidence-based screening measures should be reached.

## 6. Racial and Ethnic Variations in Survival in EO-CRC

Given the rising incidence of EO-CRC, it is important to understand the distribution patterns among various racial and ethnic groups. In a recent population-based cohort study of 22,834 patients with EO-CRC, there was a higher likelihood of EO-CRC mortality among Native Hawaiian, Other Pacific Islander and Non-Hispanic Black individuals compared to Non-Hispanic White individuals [19]. Additionally, socioeconomic status was found to be associated with the greatest change in mortality differences between Native Hawaiian or Other Pacific Islander and non-Hispanic White individuals, though the effect was modest. Further, evidence shows that Hispanic individuals have rapidly rising EO-CRC rates and increased CRC-mortality risk compared to Non-Hispanic White individuals even after adjusting for tumor characteristics, but the effect attenuated after adjustment for social and neighborhood contextual factors [28]. In a study based on the SEER database, Aloysius et al. found that poverty, unemployment, and lower income were associated with worse EO-CRC-related survival outcomes in various racial and ethnic groups [28]. However, having higher education and access to commercial health insurance was shown to improve survival. Similarly, a population-based study of 45,660 patients conducted by Ko et al. found that those with the lowest neighborhood socioeconomic status were 1.13 times (95% CI 1.06–1.21) more likely to present with metastases and had lower survival compared to those with the highest neighborhood socioeconomic status [29]. In this study, Non-Hispanic Black patients were more likely to present with metastatic disease and less likely to undergo surgery for localized or regional disease and had lower survival compared to the non-Hispanic White counterparts [30].

In addition to increased mortality in Non-White patients, disparities exist among those undergoing surgical resection for EO-CRC. Among patients with EO-CRC with locoregional disease, Black and American Indian/Alaska Native patients were less likely to undergo surgical resection, demonstrating less than half the odds compared to Non-Hispanic White, Asian/Pacific Islander and Hispanic patients. This sheds light on opportunities to understand disparities in care, address barriers and determine their impact on survival [30].

## 7. Call for Action for Awareness of EO-CRC and Future Directions

The incidence of EO-CRC is increasing worldwide, marking it as a rising global phenomenon, with the incidence of CRC projected to more than double by 2030. There is an urgent need for global initiatives and multi-disciplinary partnerships to identify risk factors of EO-CRC and implement effective preventive strategies that align with local preferences. We propose a call to action for increased public health policies, awareness initiatives, research, and global collaboration to help mitigate the systemic burden of EO-CRC. Given the aggressive tumor presentation of EO-CRC, impact extends to a direct loss of productive years to treatment, greater economic burden, and the overall personal impact, morbidity and mortality. Efforts to promote the benefits of and adherence to screening, including genetic profiling, should be considered for all patients to guide treatment strategies and counseling (Table 2).

### Addressing Barriers to EO-CRC Screening

Several studies have highlighted the racial, ethnic, and socioeconomic disparities in screening, treatment and survival among minority groups. It is imperative to understand and work to mitigate these differences to improve outcomes. Future efforts prioritizing underserved populations is necessary. We call for the action of policymakers to address structural barriers including screening access as well as collaboration efforts among clinicians, public health entities, and the community to help with interventions for improving access to screening, identification, and treatment of EO-CRC. Actions include funding public awareness campaigns, subsidizing screening initiatives among patients who are at increased risk for EO-CRC, engaging social media and technology platforms to spread awareness, and funding community health programs in underserved areas to provide information on the importance of screening measures. Additionally, it is imperative to fund epidemiological studies, research on EO-CRC risk factor identification, prevention strategies and screening of adherence rates to guide future guidelines and therapies. Further, EO-CRC-based education should be incorporated in medical training for clinicians to highlight better identification and screening practices. We encourage clinicians to focus on obtaining accurate family histories in young adults under the age of 50 about first-degree relatives with CRC. One study reported that only 70% of primary care physicians routinely recommended CRC screening among patients with a first-degree relative with CRC [36]. Assessing and mitigating barriers to obtaining detailed family histories including time restraints, unknown family histories, or knowledge gaps in screening guidelines should be addressed.

Further research is needed to understand the underlying cause and mechanisms of EO-CRC. Work is needed to initiate earlier and comprehensive screening and monitoring and coordinated efforts are needed to understand, treat, and alleviate the widespread burden on individuals with EO-CRC. Through a combination of robust research, improvements in screening awareness and initiatives, community investment, and increased education, we can significantly improve the early detection and prevention of EO-CRC.

## Figures and Tables

**Table 1 biomedicines-13-01572-t001:** Early-onset colorectal cancer clinical and histopathological features.

Category	Characteristics in EO-CRC	Characteristics in CRC	References
Incidence and Stage at Diagnosis	Rising incidence. Diagnosed at advanced stages (stage III or IV).	Declining incidence Often can be detected at earlier stages due to widespread screening practices.	[2,22]
Symptoms	Rectal bleeding, weight loss, anemia, change in bowel habits, abdominal pain.	Similar symptoms or even asymptomatic and detected during routine screening.	[2]
Histopathology	Aggressive histological subtypes including poorly differentiated tumors, mucinous carcinomas, and signet ring cell carcinomas.	Moderately differentiated adenocarcinoma, signet ring and mucinous pathology.	[20,21]
EO-CRC Oncogenic Mutations	Hereditary syndromes. Microsatellite instability and epigenetic interactions. Higher prevalence of *KRAS*, *BRAF*, and *NRAS.*	Sporadic mutations are more prevalent.	[20,21]

**Table 2 biomedicines-13-01572-t002:** Early-onset colorectal cancer characteristics and research gaps.

Category	Characteristics	Research Gaps	References
Epidemiology	Rising incidence particularly in developed countries. Highest increase in Non-Hispanic Black and Hispanic populations in the United States of America. Late presentation.	Understanding EO-CRC trends across regions, especially in low- and middle-income countries, may increase awareness of disparities. Lack of longitudinal studies on EO-CRC incidence by socioeconomic and racial/ethnic groups and survival.	[7,31]
Risk Factors	Genetic predispositions such as Lynch syndrome and familial adenomatous polyposis. Modifiable risk factors including high processed meat intake, alcohol use, tobacco use disorder, and low physical activity.	Interaction between genetic predispositions and modifiable lifestyle factors is poorly understood. Studies on how gut microbiome may impact risk may increase understanding of dietary impacts on incurring EO-CRC.	[11]
Screening	Several current society guidelines recommend starting colorectal cancer screening at age 45, reduced from age 50. Colonoscopy remains the gold standard for detecting EO-CRC.	Limited adherence to screening guidelines among younger populations. Significant disparities in screening among racial minorities and patients with low socioeconomic status. Future research and endeavors should be aimed at raising awareness and investigating barriers to seeking attention among patients with high-risk factors.	[32]
Environmental andGenetic Precursors	EO-CRC shows unique molecular characteristics, including higher prevalence of microsatellite instability, CpG island methylator phenotype, and distinct mutations (i.e., BRAF). Carcinogens associated with cigarette smoke, such as nitrosamines, benzene, heterocyclic amines, and polycyclic aromatic hydrocarbons	More studies are needed to understand the role of genetic precursors and EO-CRC. More studies are needed to understand how genetic and molecular roles may influence response to treatments and prognosis.	[13,33]
Symptoms &Diagnosis	Common symptoms include rectal bleeding, changes in bowel habits, and unintentional weight loss. Younger patients may face delayed diagnoses due to misattribution of symptoms to benign conditions. Diagnosis frequently occurs at advanced stages, leading to poorer outcomes with delays in treatment initiation.	Need for attention to symptom recognition and triage in primary care in younger patients. Development of risk calculation models for younger individuals to help stratify high-risk individuals.	[2,4]
Treatment	Survival outcomes vary based on genetic factors, age, disease severity. Young patients may be more likely to receive aggressive surgical and systemic treatments. Treatment-related side effects (i.e., infertility, mental health) disproportionately impact younger patients.	Need for comprehensive data on the longitudinal efficacy of treatment regimens in EO-CRC. Need for strategies to mitigate treatment-related quality of life impairments, including fertility preservation and mental health support.	[34,35]
Disparities	Socioeconomic factors may significantly influence EO-CRC outcomes, with low socioeconomic backgrounds experiencing higher mortality rates and decreased surgical interventions. Racial and ethnic minorities may face more barriers to diagnosis, screening, and access to advanced treatments.	Need for root-cause analysis on ethnic disparities for mortality, advocacy, and interventions that address socioeconomic challenges.	[19,29]

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
