# Peer review of "Early-Onset Colorectal Cancer: A Review of Current Insights and a Call for Action"

_biomedicines, 2025, doi:10.3390/biomedicines13071572_

Round 1
Reviewer 1 Report
Comments and Suggestions for Authors
This article reported that the rising incidence of early-onset colorectal cancer (EO-CRC) in individuals under 50 presents a growing public health concern, distinct from trends in older populations. This review highlights EO-CRC’s unique epidemiological, genetic, and clinical features, emphasizing the need for tailored prevention and management strategies. The article is arranged very poorly, and specific comments are as follows.
- Lines 13- 15: " While the incidence of colorectal cancer (CRC) in patients above .................has been increasing, posing a public health challenge" This statement is very lengthy, and should be concise for better understanding.
- Lines 21-23:" Keywords: keyword 1; keyword 2; keyword 3 (List three to ten pertinent keywords spe- 21 cific to the article yet reasonably common within the subject discipline.)" What is this?
- The introduction section of the article is very brief and should be elaborate and restructured. What was of the hypothesis of this article should be clearly mentioned in the last paragraph of the introduction.
- Lines 48-50:" However, a recent 2025 population..........America and the Caribbean” This statement is very general and should be revised.
- Lines 56-60: "These trends are notable.............cross successive generations" This statement should be cited with references.
- "4. Clinical and Histopathological Characteristics of EO-CRC" better summarized such information in Table.
- "5. Screening" This heading is very simple and should be revised.
- "Headings 7 Call for Action" should be revised and different from the title.
Author Response
- Lines 13- 15: " While the incidence of colorectal cancer (CRC) in patients above .................has been increasing, posing a public health challenge" This statement is very lengthy, and should be concise for better understanding.
Author Response: Thank you for pointing this out, we agree with your comments. We have truncated this sentence for clarity and have removed sevreal words which now makes the sentence easier to understand. Lines 13-15. - Lines 21-23:" Keywords: keyword 1; keyword 2; keyword 3 (List three to ten pertinent keywords spe- 21 cific to the article yet reasonably common within the subject discipline.)" What is this?
Author Response: Thank you for pointing this out, we agree with your comments. We have added keywords pertinent to the review article: early-onset colorectal cancer, colorectal cancer, epidemiology, microsatellite instability, modifiable risk factors. Lines 20-22. - The introduction section of the article is very brief and should be elaborate and restructured. What was of the hypothesis of this article should be clearly mentioned in the last paragraph of the introduction.
Author Response: Thank you for pointing this out, we agree with your comments. We have elaborated more in the introduction to better outline the framework of the review article. We also added a hypothesis in the last paragraph of the introduction. Lines 30-35, 38-41, 55-58. - Lines 48-50:" However, a recent 2025 population..........America and the Caribbean” This statement is very general and should be revised.
Author Response: Thank you for pointing this out, we agree with your comments. We have revised this statement and elaborated on this point and made it more clear by highlighting that EO-CRC is not only limited to developing countries but is a global issue. Lines 57-59. - Lines 56-60: "These trends are notable.............cross successive generations" This statement should be cited with references.
Author Response: Thank you for pointing this out, we agree with your comments. This statement was cited with the appropriate reference. Line 59. - "4. Clinical and Histopathological Characteristics of EO-CRC" better summarized such information in Table.
Author Response: Thank you for pointing this out, we agree with your comments. We have added an additional table outlining clinical and histopathological characteristics in EO-CRC in Table 1. Line 236, Table 1. - "5. Screening" This heading is very simple and should be revised.
Author Response: Thank you for pointing this out, we agree with your comments. We revised the title to specify the section is about screening in EO-CRC. Line 217. - "Headings 7 Call for Action" should be revised and different from the title.
Author Response: Thank you for pointing this out, we agree with your comments. We have revised this section heading to be more specific and read as "Call for Action For Awareness to EO-CRC and Future Directions". Line 279.

Reviewer 2 Report
Comments and Suggestions for Authors
The review leads us to a more and more important public health problem, EO-CR which is on the rise in terms of global incidence despite improved screening in elderly populations. This makes the topic both timely and clinically relevant. i do have some concern here:
- The flow and conciseness are impacted by the needless repetition of some points . pls check throughout the manuscript.
- Uncertain alignment of specific data to cited studies, references like "13,15" with unclear placement, and inconsistent citation formatting (such as using "3-4" instead of "3,4"), can all lower credibility and traceability.
- Though it speaks of the purpose of providing "current insights" (lines 33–34), it does not clearly state what this review is going to add uniquely or why it's needed today.
- A review introduction must, establish background, inform us about why the topic is significant today, identify knowledge gaps or controversies and state the purpose and scope of the review. hencer pls revise your introduction
- There are a number of awkward or inaccurate phrases (I see this throughout the manuscript)
- The shift between topics (e.g., from global incidence trends to mortality disparities) is abrupt. More transitional sentences would improve logical flow.
- For a data-heavy section, a figure or table showing global EO-CRC incidence trends or birth cohort data would enhance comprehension.
-
Insufficient Introduction to Section 4
-
Consider adding a brief introductory paragraph to Section 4 that outlines the importance of understanding the clinical, molecular, and histopathological features of EO-CRC, especially in comparison to late-onset CRC. This will help orient the reader before diving into the details.
-
Revise the subsection titles under Section 4 to better reflect their contents. For example, use “Clinical Presentation and Diagnostic Delays” and “Molecular and Histopathological Features” to improve clarity and logical flow.
-
While the section presents important data, it reads mostly as a summary of studies. Consider incorporating interpretive commentary to help the reader understand the clinical and research implications of the findings (e.g., why TP53 mutations or signet ring histology matter in prognosis or treatment).
-
The issue of delayed diagnosis in EO-CRC is repeated multiple times. Consider consolidating these points into one or two concise, impactful paragraphs to improve readability.
-
The screening section presents current recommendations clearly, but it would benefit from a discussion of practical challenges in implementing earlier screening (e.g., healthcare resource limitations, risk of overdiagnosis, or screening uptake in younger populations).
-
Consider adding brief transition sentences to connect major subsections and help the reader follow the progression from clinical presentation to molecular pathology to screening.
-
A short conclusion at the end of Section 4 summarizing key takeaways (e.g., EO-CRC presents uniquely and requires tailored diagnostic and treatment approaches) would improve cohesion and reinforce the section’s message.
- The section would be stronger with a more interesting opening that declares the worldwide importance of EO-CRC and the need for a call to action. Attempt rewriting the opening paragraph to make it clearer why EO-CRC is a public health priority
- his section switches back and forth between topics (screening, disparities, molecular biology, clinician behavior) quite abruptly at the moment. Reorder or consolidate material under thematic subheadings or at least use transitional phrases between paragraphs to improve flow and readability.
- Table 1 is helpful, but the format appears crowded and possibly hard to read. Rewrite for better alignment and readability. Also, ensure that all studies mentioned are cited consistently and listed in the list of references.
- The table lists important research gaps but does not attempt any prioritization or strategic direction. Some commentary regarding which gaps are most important or feasible to address may give focus to future research endeavors.
- he physician behavior and family history section is important but a bit hidden. Attempt to bring it more to the forefront or make it a subsection of its own with more detail on how closing this gap would improve early detection.
- he call to action to policymakers is generic. Consider specifying exact actions (e.g., subsidizing screening for high-risk groups, funding public awareness campaigns, implementing guideline reminders in electronic health records).
- Roles of clinician and policymaker are addressed, but the role of patient advocacy organizations or public awareness campaigns is not developed. Including these would strengthen the multifaceted nature of the call for action.
- he section ends somewhat abruptly. Consider adding a short, punchy concluding paragraph that summarizes the urgent needs and emphasizes the cross-research, clinical care, and public health collaborative effort required.
Author Response
Comments and Suggestions for Authors
The review leads us to a more and more important public health problem, EO-CR which is on the rise in terms of global incidence despite improved screening in elderly populations. This makes the topic both timely and clinically relevant. i do have some concern here:
- The flow and conciseness are impacted by the needless repetition of some points . pls check throughout the manuscript.
- Author Response: Thank you for your comments, we agree. Several areas throughout the manuscript have been edited to improve flow, repetition and conciseness. Section 3.2 Lines 458-560 have been removed to reduce repetition and to increase clarity. Section 4.2 lines 586-588 have been removed to reduce repetition. Section 6 Lines 741-746 have been deleted to reduce redundant information.
- Uncertain alignment of specific data to cited studies, references like "13,15" with unclear placement, and inconsistent citation formatting (such as using "3-4" instead of "3,4"), can all lower credibility and traceability.
- Author Response: Thank you for your comments, we agree. References format have been edited. Line 29.
- Though it speaks of the purpose of providing "current insights" (lines 33–34), it does not clearly state what this review is going to add uniquely or why it's needed today.
- Author Response: Thank you for your comments, we agree. Introduction has been expanded to encompass more details regarding what has been known and why it is important to highlight knowledge on EO-CRC. Lines 29-58.
- A review introduction must, establish background, inform us about why the topic is significant today, identify knowledge gaps or controversies and state the purpose and scope of the review. hencer pls revise your introduction
- Author Response: Thank you for your comments, we agree. Introduction has been expanded to encompass more details regarding what has been known and why it is important to highlight knowledge on EO-CRC. Lines 29-58.
- There are a number of awkward or inaccurate phrases (I see this throughout the manuscript)
- Author Response: Thank you for your comments, we agree. Phrases in Lines 88-89 have been edited to be more cohesive and improve the flow of the manuscript.
- The shift between topics (e.g., from global incidence trends to mortality disparities) is abrupt. More transitional sentences would improve logical flow.
- Author Response: Thank you for your comments, we agree. We have added various subsections in Section 2 to provide better organization and flow between epidemiology to mortality disparities. Lines 92-118.
- For a data-heavy section, a figure or table showing global EO-CRC incidence trends or birth cohort data would enhance comprehension.
- Author Response: Thank you for your comments, we agree. We have clarified this section to make it more clear to understand by increasing clarity and describing trends.
Insufficient Introduction to Section 4
- Author Response: Thank you for your comments, we agree. A transition sentence in Section 4 Lines 188-190 has been added to introduce the histopathological and clinical features of EO-CRC.
- Consider adding a brief introductory paragraph to Section 4 that outlines the importance of understanding the clinical, molecular, and histopathological features of EO-CRC, especially in comparison to late-onset CRC. This will help orient the reader before diving into the details.
- Author Response: Thank you for your comments, we agree. A transition sentence in Section 4 Lines 188-190 has been added to introduce the histopathological and clinical features of EO-CRC.
- Revise the subsection titles under Section 4 to better reflect their contents. For example, use “Clinical Presentation and Diagnostic Delays” and “Molecular and Histopathological Features” to improve clarity and logical flow.
- Author Response: Thank you for your comments, we agree. The subsections are added within Section 4 to improve clarity and logical flow. Lines 195 and 222.
- While the section presents important data, it reads mostly as a summary of studies. Consider incorporating interpretive commentary to help the reader understand the clinical and research implications of the findings (e.g., why TP53 mutations or signet ring histology matter in prognosis or treatment).
- Author Response: Thank you for your comments, we agree. We have added more detail about prognosis and treatment in Lines 574-576.
- The issue of delayed diagnosis in EO-CRC is repeated multiple times. Consider consolidating these points into one or two concise, impactful paragraphs to improve readability.
- Author Response: Thank you for your comments, we agree. Delays in diagnosis in EO-CRC are mainly consolidated in Section 4.1 and in Table 2. Lines 196-222.
- The screening section presents current recommendations clearly, but it would benefit from a discussion of practical challenges in implementing earlier screening (e.g., healthcare resource limitations, risk of overdiagnosis, or screening uptake in younger populations).
- Author Response: Thank you for your comments, we agree. Barriers to implementing earlier screening have been described in lines 282-286.
- Consider adding brief transition sentences to connect major subsections and help the reader follow the progression from clinical presentation to molecular pathology to screening.
- Author Response: Thank you for your comments, we agree. We added a brief transition section to bridge subsections 4.1 and 4.2 within line 225-226.
- A short conclusion at the end of Section 4 summarizing key takeaways (e.g., EO-CRC presents uniquely and requires tailored diagnostic and treatment approaches) would improve cohesion and reinforce the section’s message.
- Author Response: Thank you for your comments, we agree. We added a brief closing sentence describing key takeaways in lines 249-252.
- The section would be stronger with a more interesting opening that declares the worldwide importance of EO-CRC and the need for a call to action. Attempt rewriting the opening paragraph to make it clearer why EO-CRC is a public health priority
- Author Response: Thank you for your comments, we agree. We have edited the introduction of the first section to be stronger and more interesting by highlighting its rising prevalence, and need for urgent global initiatives. We also discuss a proposal to the call for action. Lines 328-334. We discuss further public health proposals and ways to spread awareness throughout this section.
- This section switches back and forth between topics (screening, disparities, molecular biology, clinician behavior) quite abruptly at the moment. Reorder or consolidate material under thematic subheadings or at least use transitional phrases between paragraphs to improve flow and readability.
- Author Response: Thank you for your comments, we agree. Several sub sections have been made throughout the manuscript and transitional phases between paragraphs have been inserted to improve flow.
- Table 1 is helpful, but the format appears crowded and possibly hard to read. Rewrite for better alignment and readability. Also, ensure that all studies mentioned are cited consistently and listed in the list of references.
- Author Response: Thank you for your comments, we agree. Table 1 formatting has been edited to improve alignment and readability and references have been verified.
- The table lists important research gaps but does not attempt any prioritization or strategic direction. Some commentary regarding which gaps are most important or feasible to address may give focus to future research endeavors.
- Author Response: Thank you for your comments, we agree. We have significantly edited Table 1 to include more information about feasibility and future directions of research throughout the table.
- he physician behavior and family history section is important but a bit hidden. Attempt to bring it more to the forefront or make it a subsection of its own with more detail on how closing this gap would improve early detection.
- Author Response: Thank you for your comments, we agree. We have added subsection 7.1 titled “addressing barriers to EO-CRC Screening” in Line 336.
- The call to action to policymakers is generic. Consider specifying exact actions (e.g., subsidizing screening for high-risk groups, funding public awareness campaigns, implementing guideline reminders in electronic health records).
- Author Response: Thank you for your comments, we agree. We have added several calls to action for policymakers and public health initiatives in Section 7.1. Lines 346-353.
- Roles of clinician and policymaker are addressed, but the role of patient advocacy organizations or public awareness campaigns is not developed. Including these would strengthen the multifaceted nature of the call for action.
- Author Response: Thank you for your comments, we agree. We have added several calls to action for policymakers and public health initiatives in Section 7.1, specifically about funding community health programs, engaging social media and technology platforms to raise awareness, and the need for further epidemiological studies on this topic. Lines 346-353.
- he section ends somewhat abruptly. Consider adding a short, punchy concluding paragraph that summarizes the urgent needs and emphasizes the cross-research, clinical care, and public health collaborative effort required.
- Author Response: Thank you for your comments, we agree. We have added a strong concluding sentence in Lines 364-366 to emphasize public health collaboration, research, and clinical care in order to help with early detection and prevention of EO-CRC. Lines 364-466.

Round 2
Reviewer 1 Report
Comments and Suggestions for Authors
The carefully response the raised comment and suggestion, therefore, I would recommend the article for publication.
Reviewer 2 Report
Comments and Suggestions for Authors
Thank you for answering my question. Good luck